# Distinct roles for extracellular and intracellular domains in neuroligin function at inhibitory synapses

**Quynh-Anh Nguyen[1,2], Meryl E Horn[1,2], Roger A Nicoll[1,3]***

[1]Department of Cellular and Molecular Pharmacology, University of California, San Francisco, San Francisco, United States; [2]Neuroscience Graduate Program, University of California, San Francisco, San Francisco, United States; [3]Department of Physiology, University of California, San Francisco, San Francisco, United States

**Abstract** Neuroligins (NLGNs) are postsynaptic cell adhesion molecules that interact trans-synaptically with neurexins to mediate synapse development and function. NLGN2 is only at inhibitory synapses while NLGN3 is at both excitatory and inhibitory synapses. We found that NLGN3 function at inhibitory synapses in rat CA1 depends on the presence of NLGN2 and identified a domain in the extracellular region that accounted for this functional difference between NLGN2 and 3 specifically at inhibitory synapses. We further show that the presence of a cytoplasmic tail (c-tail) is indispensible, and identified two domains in the c-tail that are necessary for NLGN function at inhibitory synapses. These domains point to a gephyrin-dependent mechanism that is disrupted by an autism-associated mutation at R705 and a gephyrin-independent mechanism reliant on a putative phosphorylation site at S714. Our work highlights unique and separate roles for the extracellular and intracellular regions in specifying and carrying out NLGN function respectively.

**\*For correspondence:** roger. nicoll@ucsf.edu

**Competing interests:** The authors declare that no competing interests exist.

## Introduction

Proper balance between inhibitory and excitatory connections is important for proper functioning of neuronal circuits (*Maćkowiak et al., 2014*; *Bang and Owczarek, 2013*). Cell adhesion molecules have a critical role in coordinating the apposition of presynaptic terminals with postsynaptic sites of differentiation (*Washbourne et al., 2004*; *Yamagata et al., 2003*). While much work has been done to determine the molecular organization of cell adhesion molecules at excitatory synapses, less is known about how these components are organized at inhibitory synapses.

Neuroligins are a family of postsynaptic cell adhesion molecules that interact trans-synaptically with their corresponding presynaptic neurexin binding partners to mediate proper synaptic function (*Südhof, 2008*; *Chih et al., 2005*). Rats express three neuroligins (NLGN1-3): NLGN1 is expressed exclusively at excitatory synapses, NLGN2 is selectively present at inhibitory synapses, while NLGN3 is found at both excitatory and inhibitory synapses (*Varoqueaux et al., 2004*; *Budreck and Scheif-fele, 2007*; *Song et al., 1999*). NLGN2 and 3 share many previously identified domains both in their extracellular and intracellular regions. Despite the similarities between the two proteins, it is currently unknown whether they perform the same function at inhibitory synapses. One notable difference between NLGN2 and 3 resides in the extracellular region at splice site A which has previously been proposed to affect NLGN binding to its presynaptic neurexin partner (*Ichtchenko et al., 1996*; *Chih et al., 2006*).

It has been suggested that NLGN2 functions through a direct interaction on its cytoplasmic tail with gephyrin, a scaffold protein thought to be essential for stabilizing glycine and GABA$_A$ receptors

at inhibitory synapses (*Choii and Ko, 2015*; *Tyagarajan and Fritschy, 2014*). In addition, it has been proposed that collybistin, a brain-specific guanine nucleotide exchange factor (GEF), helps regulate the localization of gephyrin, and that NLGN2 is a specific activator of collybistin via a direct interaction at the proline rich region in its cytoplasmic tail (*Kins et al., 2000*; *Poulopoulos et al., 2009*; *Soykan et al., 2014*). While previous studies have been able to link these interactions to NLGN2, there has been no direct study of whether these interactions are necessary for proper functioning of NLGNs at inhibitory synapses.

To investigate the importance of NLGN2 and 3 at inhibitory synapses, we used microRNAs targeted to NLGN2 or 3 individually. We found that NLGN2 is a critical component of inhibitory synapses while NLGN3 function at inhibitory synapses depends on the presence of NLGN2. Further investigation expressing chimeric constructs of NLGN2 and 3 in isolation, using a microRNA targeting all three endogenously expressed NLGNs 1, 2, and 3, identified a previously uncharacterized domain in the extracellular region that accounted for this functional difference between NLGN2 and 3. Using a similar technique to study the importance of the intracellular region in NLGN function at inhibitory synapses, we found a critical requirement for the cytoplasmic tail and identified two key residues that are separately involved in gephyrin-dependent and gephyrin-independent mechanisms of NLGN function at inhibitory synapses. We further show that an autism-associated mutation inhibits the gephyrin-dependent pathway while a phosphorylation site is responsible for modulating the gephyrin-independent pathway. These findings identify new mechanisms for NLGN function, particularly at inhibitory synapses, and provide new avenues of study for elucidating the molecular mechanisms present at inhibitory synapses.

## Results

### NLGN2 is the critical NLGN at inhibitory synapses

To determine the relative contributions of NLGN2 and 3 to inhibitory synaptic transmission, we utilized targeted microRNA constructs to knockdown either NLGN2 or 3 (validated in *Figure 1—figure supplement 1a* and *Shipman and Nicoll, 2012a*). We biolistically transfected organotypic hippocampal slices with our constructs of interest and performed dual-whole cell recordings from CA1 neurons 7–10 days after transfection. Compared to a previously validated knockdown construct of NLGNs 1–3 (*Shipman et al., 2011*) which reduces inhibitory synaptic transmission to about 50% (*Figure 1a*) we found that while NLGN2 knockdown alone recapitulated the 50% decrease in inhibitory synaptic transmission (*Figure 1b and d*), NLGN3 knockdown alone had no effect on inhibitory synaptic transmission (*Figure 1c and d*). Furthermore, while overexpression of NLGN3 alone does enhance inhibitory synaptic transmission (*Figure 1e and g*), overexpression of NLGN3 with a NLGN2 knockdown construct fails to enhance inhibitory responses (*Figure 1f and g*), suggesting that NLGN3 requires the presence of NLGN2 to function at inhibitory synapses. This is consistent with previous results showing no enhancement of inhibitory responses when NLGN3 was overexpressed on a NLGN1-3 knockdown background (*Shipman et al., 2011*). Notably, NLGN3 overexpression on a NLGN2 knockdown background still enhanced excitatory responses similar to NLGN3 overexpression alone (*Figure 1—figure supplement 1b,c and d*), suggesting our effects are specific for inhibitory synapses.

### Difference between NLGN 2 and 3 function at inhibitory synapses resides in distinct domain in the extracellular region

To determine where this functional difference between NLGN2 and 3 at inhibitory synapses resides, we made chimeric constructs and expressed them on the reduced endogenous NLGN (NLGN1-3miR) background to study the effects of our constructs in isolation (*Figure 2a*). Expression of the full-length NLGN2 resulted in a large enhancement of inhibitory currents (*Figure 2b and e*). Expression of a construct containing the NLGN2 extracellular region and NLGN3 cytoplasmic tail enhanced inhibitory currents (*Figure 2c and e*) similar to full-length NLGN2 (*Figure 2e*). However, expression of a construct containing the NLGN3 extracellular region and NLGN2 intracellular region failed to enhance inhibitory currents (*Figure 2d and e*), suggesting that the functional difference between NLGN2 and 3 at inhibitory synapses resides in the extracellular region. Importantly, while the NLGN3 extracellular-NLGN2 intracellular chimeric construct failed to enhance inhibitory responses,

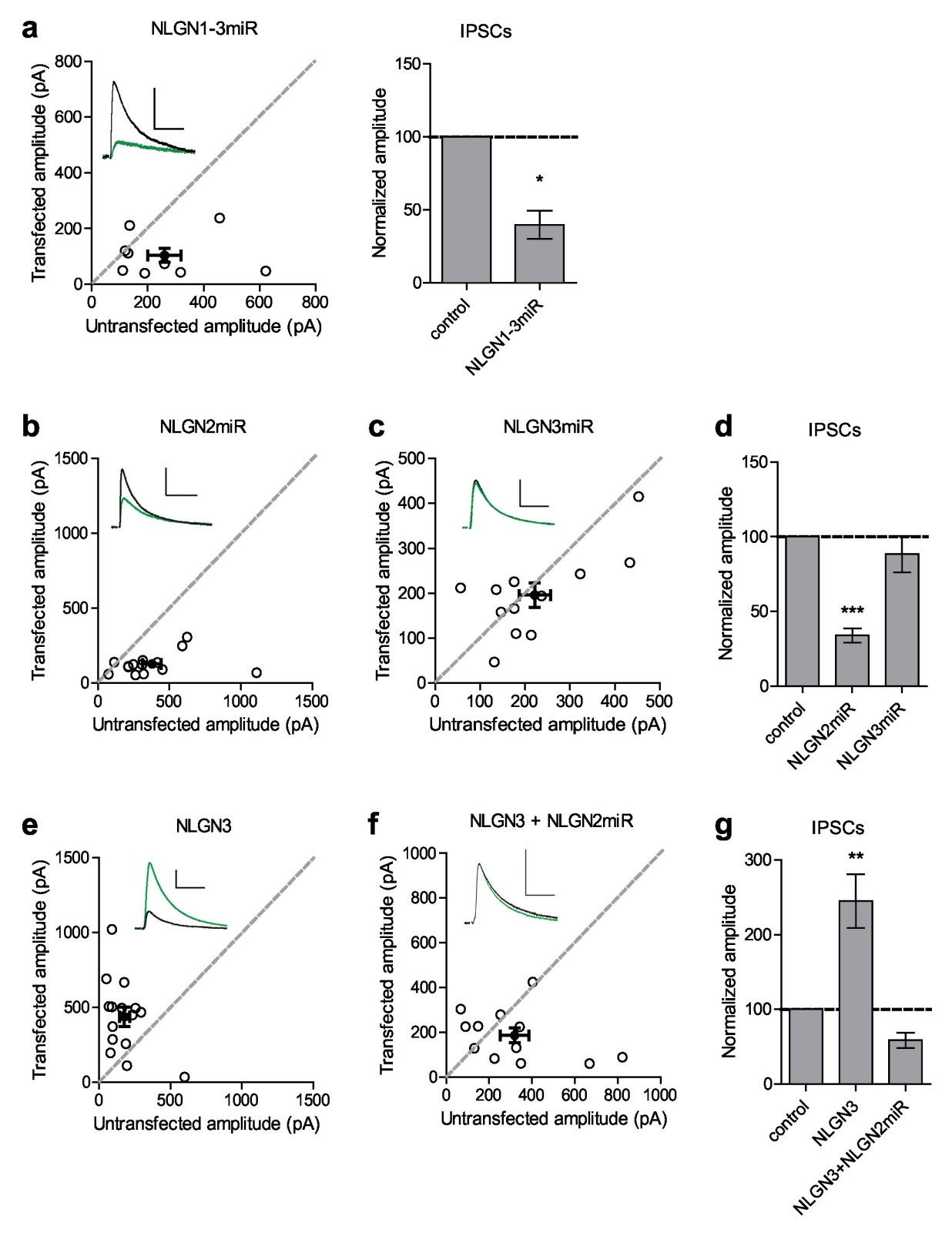

**Figure 1.** Neuroligin 2 is the critical neuroligin at inhibitory synapses. (**a**) Knockdown of endogenous neuroligins 1–3 reduces inhibitory responses compared to untransfected control cells (*p=0.0391, n = 9). (**b**) Scatter plot showing reduction in IPSCs in NLGN2 microRNA-transfected neurons compared to untransfected controls (***p=0.0002, n = 15). (**c**) Scatter plot showing no reduction in IPSCs in NLGN3 microRNA-transfected neurons compared to untransfected controls (p=0.3013, n = 12). (**d**) Summary graph of b and c. (**e**) Scatter plot showing overexpression of NLGN3 enhances

*Figure 1 continued on next page*

*Figure 1 continued*

IPSCs compared to untransfected controls (**p=0.0084, n = 15). (**f**) Scatter plot showing overexpression of NLGN3 combined with NLGN2 knockdown fails to enhance IPSCs (p=0.2661, n = 12). (**g**) Summary graph of e and f. For panels a-c and e-f, open circles are individual pairs, filled circle is mean ± s. e.m. Black sample traces are control, green are transfected. Scale bars represent 100 pA and 50 ms. For panels a, d, and g summary graph plots mean transfected amplitude ± s.e.m, expressed as a percentage of control amplitude. Significance above each column represents pairwise comparison between transfected and untransfected cells. See also *Figure 1—figure supplement 1*.

The following figure supplement is available for figure 1:

**Figure supplement 1.** Validation of NLGN2 knockdown construct and EPSCs for NLGN3 overexpression.

---

it was still able to potentiate excitatory responses (*Figure 2—figure supplement 1*), indicating that this construct is expressed and functional. It further shows that there are differential requirements for NLGN3 function at inhibitory versus excitatory synapses, and that the NLGN2 c-tail is capable of functioning at excitatory synapses.

We constructed further chimeric constructs to determine where in the extracellular region this difference between NLGN2 and 3 resides and if we could effectively confer NLGN2 function onto NLGN3 by transplanting a critical domain. There are various components located in the extracellular region which have previously been suggested to be important for NLGN function (*Figure 3a* and *Figure 3—figure supplement 1a*). One key component of the extracellular region, the critical dimerization residues, are conserved between NLGN2 and 3, suggesting that differences between NLGN2 and 3 may not be due to differences in ability to dimerize (*Dean et al., 2003*; *Ko et al., 2009*; *Shipman and Nicoll, 2012b*). One notable domain of interest resides in splice site A, which is important for neurexin binding (*Ichtchenko et al., 1996*; *Chih et al., 2006*). Interestingly, there are many differences between NLGN2 and 3 at this splice site, and these might account for the functional differences between the two.

Starting from the proximal region closest to the transmembrane region, we progressively added back regions of NLGN2 onto the NLGN3 extracellular region while keeping the NLGN2 c-tail intact. We first inserted a region of NLGN2 up to the 623rd amino acid that contains substantial differences between it and NLGN3 (*Figure 3a* and *Figure 3—figure supplement 1a*). Neither this chimeric construct (*Figure 3b and h*) nor one adding back NLGN2 right before splice site A at the 180th amino acid (*Figure 3c and h*) was able to enhance inhibitory currents. However, adding back NLGN2 up to the 52nd amino acid was able to potentiate inhibitory currents (*Figure 3d and h*). Further refinement showed that transplanting just the domain between the 52nd and 180th amino acid of NLGN2 was able to fully confer NLGN2 ability to potentiate inhibitory currents onto NLGN3 (*Figure 3e and h*). Since this domain encompassed the splice site A and a previously uncharacterized domain of the extracellular region, we wanted to know whether the splice site A or the uncharacterized domain alone was sufficient to confer NLGN2 function onto NLGN3. While expression of a chimeric construct containing the uncharacterized extracellular domain of NLGN2 between amino acids 52–164 showed a modest enhancement of inhibitory currents (*Figure 3f and h*), it was far less than the effect of the full length NLGN2 (*Figure 3h*). A chimera containing only the splice site A domain of NLGN2 (*Figure 3g and h*) failed to enhance inhibitory responses. Importantly, all chimeric constructs, except NLGN3-180-NLGN2 which appears non-functional, that failed to potentiate inhibitory responses were still able to potentiate excitatory responses, indicating that they were functional (*Figure 3—figure supplement 1b*). Together, this suggests that the splice site A domain alone does not account for the differences between NLGN2 and 3 function but that a previously uncharacterized adjacent domain is also required.

## A c-tail is required for NLGN function

Since the c-tails of the NLGNs share many similar domains (*Figure 4a*), and our previous results showed that the c-tails of NLGN2 and 3 were interchangeable (*Figure 2c and d*), we wanted to know whether the c-tail of NLGN1, a NLGN that is selectively present at excitatory synapses, was also capable of functioning at inhibitory synapses. In experiments done on a reduced endogenous NLGN background, full-length NLGN1 was unable to enhance inhibitory responses (*Figure 4b and d*), while a chimeric construct containing the NLGN2 extracellular region with a NLGN1 intracellular

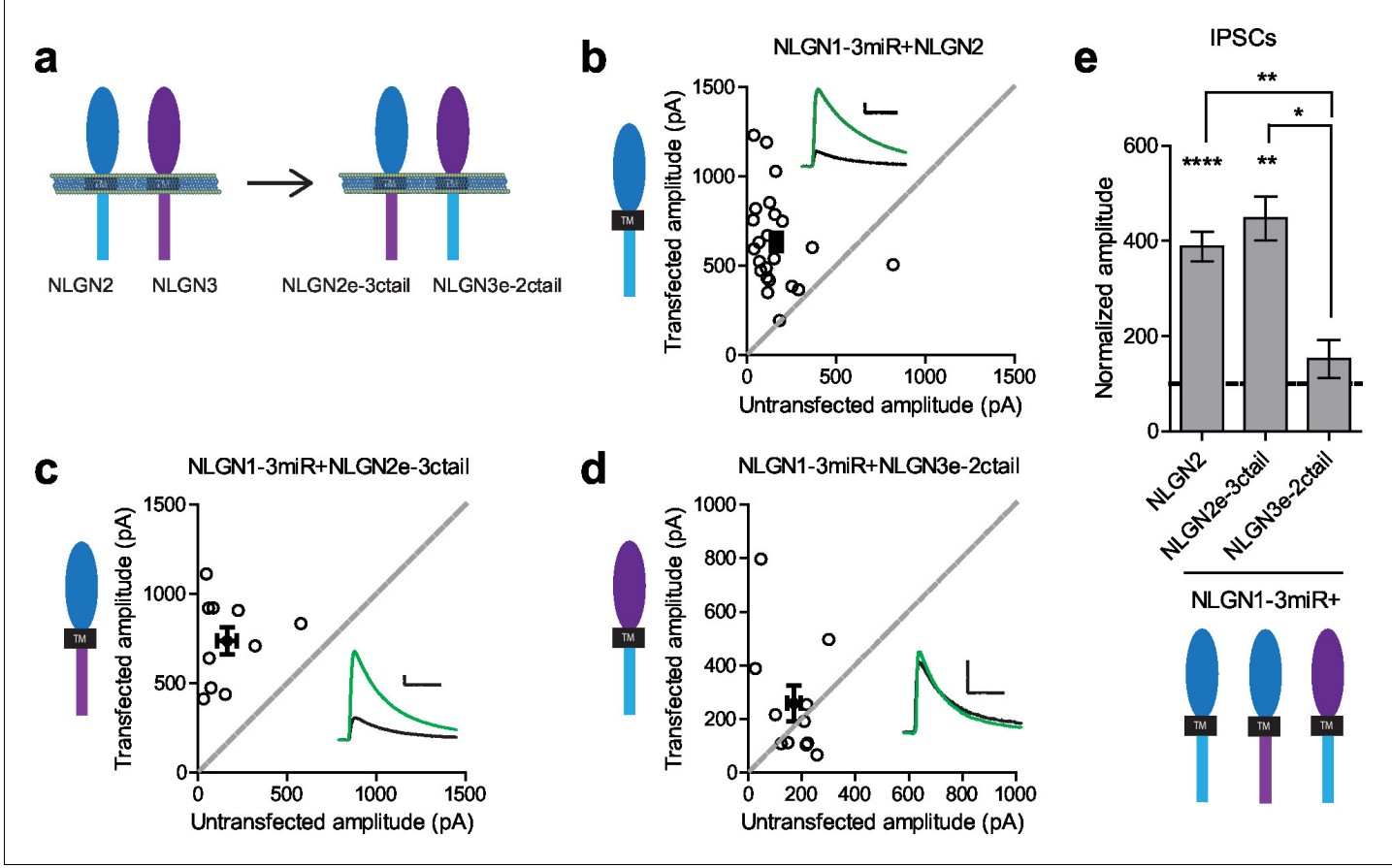

**Figure 2.** Difference between NLGN2 and 3 function at inhibitory synapses resides in the extracellular region. (**a**) To determine where this functional difference between NLGN2 and 3 resides, we made chimeric constructs and expressed them on a reduced endogenous neuroligin background. Black TM is transmembrane domain. (**b**) Scatter plot showing expression of NLGN2 on a NLGN1-3miR knockdown background enhances IPSCs compared to untransfected controls (****p<0.0001, n = 25). (**c**) Scatter plot showing expression of a construct containing the NLGN2 extracellular region and NLGN3 cytoplasmic tail (NLGN2e-3ctail) on a NLGN1-3miR knockdown background enhanced IPSCs (**p=0.002, n = 10). (**d**) Scatter plot showing expression of a construct containing the NLGN3 extracellular region and NLGN2 cytoplasmic tail (NLGN3e-2ctail) on a NLGN1-3miR knockdown background failed to enhance IPSCs (p=0.5771, n = 11). (**e**) Summary graph of b-d. Effect of expressing NLGN3e-2ctail is significantly less than expression of NLGN2e-3ctail (*p=0.0124) or NLGN2 (**p=0.0067). For panels b-d, open circles are individual pairs, filled circle is mean ± s.e.m. Black sample traces are control, green are transfected. Scale bars represent 100 pA and 50 ms. For panel e, graph plots mean transfected amplitude ± s.e.m, expressed as a percentage of control amplitude. Significance above each column represents pairwise comparison between transfected and untransfected cells. See also *Figure 2— figure supplement 1*.

The following figure supplement is available for figure 2:

**Figure supplement 1.** Distinct requirements for NLGN function between inhibitory and excitatory synapses.

region was able to enhance inhibitory currents (*Figure 4c and d*). To determine whether a c-tail is required at all, we created a truncation mutant lacking most of the c-tail except for a few amino acids closest to the transmembrane domain which might be necessary for proper trafficking of the protein (Δ133, *Figure 4a*). While overexpression of this truncation mutant on a wild-type background (*Figure 4—figure supplement 1b and c*) enhanced inhibitory currents to a similar extent as full-length NLGN2 (*Figure 4—figure supplement 1a and c*), expression on a reduced endogenous NLGN background showed no potentiation (*Figure 4e and f*), indicating that a c-tail is required for proper NLGN function and further underlies the need to observe the effects of our mutant NLGNs on a reduced endogenous background due to the possibility of heterodimerization with endogenous NLGN. Importantly, the effect of the c-tail truncation was not due to impaired surface trafficking as

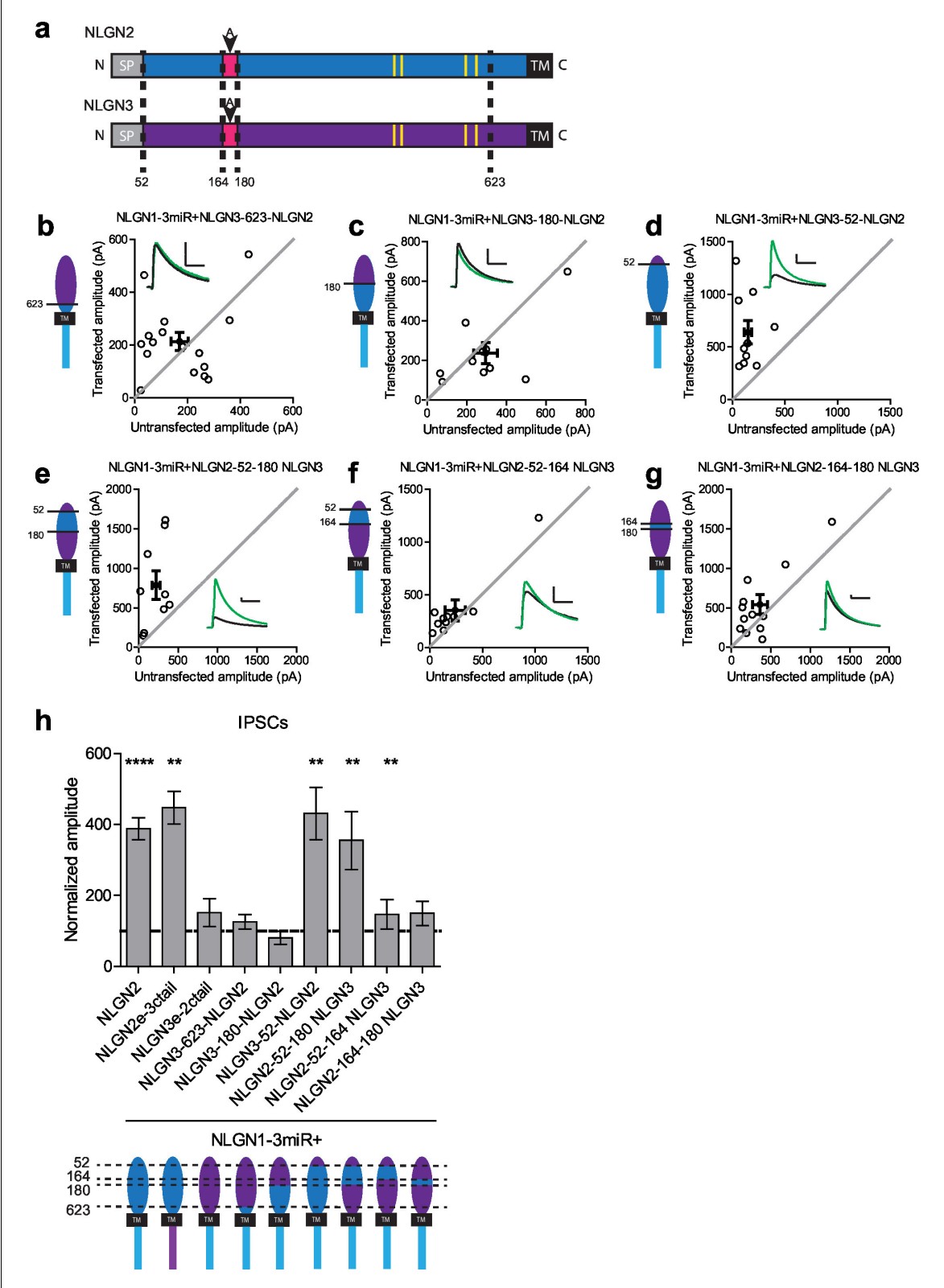

**Figure 3.** Identification of critical domain to confer NLGN function at inhibitory synapses. (a) Schematic of NLGN2 and 3 showing positions of chimera. Yellow are critical dimerization residues, pink is splice site A, grey SP is signal peptide, and black TM is transmembrane domain. (b) Scatter plot showing that adding NLGN2 onto the NLGN3 extracellular region up to the 623rd residue fails to enhance IPSCs (p=0.4229, n = 16). (c) Scatter plot showing that adding up to the 180th residue also fails to enhance IPSCs (p=0.2754, n = 10). (d) Scatter plot showing that adding up to the 52nd residue

*Figure 3 continued on next page*

*Figure 3 continued*

is successful at conferring NLGN function at inhibitory synapses and enhancing IPSCs (**p=0.002, n = 10) (e) Expression of further chimeric constructs identifies a domain between residue 52–180 on NLGN2 that is sufficient to confer the ability to enhance IPSCs to NLGN3 (**p=0.0039, n = 9). Within this region, we also expressed only (f) the region between 52–164 (**p=0.0098, n = 10) or (g) residues corresponding to differences in splice site A (164–180) (p=0.0522, n = 12). (h) Summary graph. While the chimeric construct NLGN2-52-164-NLGN3 showed enhancement of IPSCs compared to untransfected control cells, the level of potentiation was significantly less than what we see with full-length NLGN2 (*p=0.0152). For panels b-g, open circles are individual pairs, filled circle is mean ± s.e.m. Black sample traces are control, green are transfected. Scale bars represent 100 pA and 50 ms. For panel h, graph plots mean transfected amplitude ± s.e.m, expressed as a percentage of control amplitude. Significance above each column represents pairwise comparison between transfected and untransfected cells. See also *Figure 3—figure supplement 1*.

The following figure supplement is available for figure 3:

**Figure supplement 1.** Chimeric extracellular mutants and effect on excitatory transmission.

shown by surface immunostaining of the HA-tagged NLGN2 constructs (*Figure 4—figure supplement 1d and e*).

## Previously identified domains not critical for NLGN function at inhibitory synapses

To determine what domain(s) in the c-tail are necessary for NLGN function at inhibitory synapses, we made a series of truncations to the c-tail of NLGN2, deleting previously identified domains proposed to be important for its function (full diagram of relevant sites in *Figure 5—figure supplement 1*). First, it has previously been suggested that PSD-95 binds NLGN2, since it contains a PDZ domain (*Irie et al., 1997*). Second, in addition to PSD-95, which is localized to excitatory synapses, S-SCAM is another protein that is localized to both excitatory and inhibitory synapses and can also interact via the NLGN2 PDZ domain (*Woo et al., 2013*). To determine whether any of these interactions via the PDZ domain are important for NLGN2 function, we made a truncation mutant lacking the entire PDZ domain (NLGN2Δ4). We found that deleting the PDZ domain did not alter the function of NLGN2 (*Figure 5a and d*). Proximal to the PDZ domain is the proline-rich region, which has been shown to be important for collybistin binding and enabling the recruitment of gephyrin (*Kins et al., 2000*; *Poulopoulos et al., 2009*; *Soykan et al., 2014*). However, excision of this domain (NLGN2Δ32–39) or truncation of this domain and everything downstream of it including the PDZ domain (NLGN2Δ39) still resulted in significant potentiation of inhibitory currents (*Figure 5b and d*).

Another key domain suggested to be critical for NLGN function is the gephyrin-binding domain (*Poulopoulos et al., 2009*). The reigning model of NLGN function at inhibitory synapses purports that the interaction of gephyrin with NLGN2 via its gephyrin-binding domain enables the recruitment of $GABA_A$ receptors to the synapse (*Choii and Ko, 2015*; *Tyagarajan and Fritschy, 2014*). However, there have also been reports of gephyrin-independent mechanisms of $GABA_A$ receptor enrichment at inhibitory synapses (*Lévi et al., 2004*; *Kneussel et al., 2001*). To determine whether the gephyrin-binding domain is critical for NLGN2 function, we employed a variety of methods to either disrupt or excise this region of interest. Surprisingly, we found that expression of NLGN2 with a phospho-mimic mutation previously shown to disrupt gephyrin binding (NLGN2 Y770A) (*Poulopoulos et al., 2009*; *Giannone et al., 2013*), excision of the whole gephyrin-binding domain (NLGN2Δ55–69), or, most dramatically, truncation of the gephyrin-binding domain and everything downstream of it, including all known functional domains of the NLGN2 c-tail (NLGN2Δ69), all still displayed significant potentiation of inhibitory currents (*Figure 5c and d*).

While our results suggest that the gephyrin-binding domain is not critical for NLGN2 function, we wondered whether gephyrin, itself, is required. To address this, we created a targeted knockdown construct for gephyrin that effectively knocks down more than 90% of gephyrin expression (*Figure 6—figure supplement 1a and b*). Expression of this construct reduced inhibitory currents to about 50%, similar to knockdown of all endogenous NLGNs (*Figure 6a and c*). Surprisingly, a NLGN2 truncation mutant lacking all known functional domains of the c-tail (NLGN2Δ69) coexpressed with our gephyrin and endogenous NLGN knockdown constructs was fully capable of enhancing inhibitory currents (*Figure 6b and c*). These results indicate that NLGN2 can function, not only without previously known domains, but entirely independently of gephyrin as well.

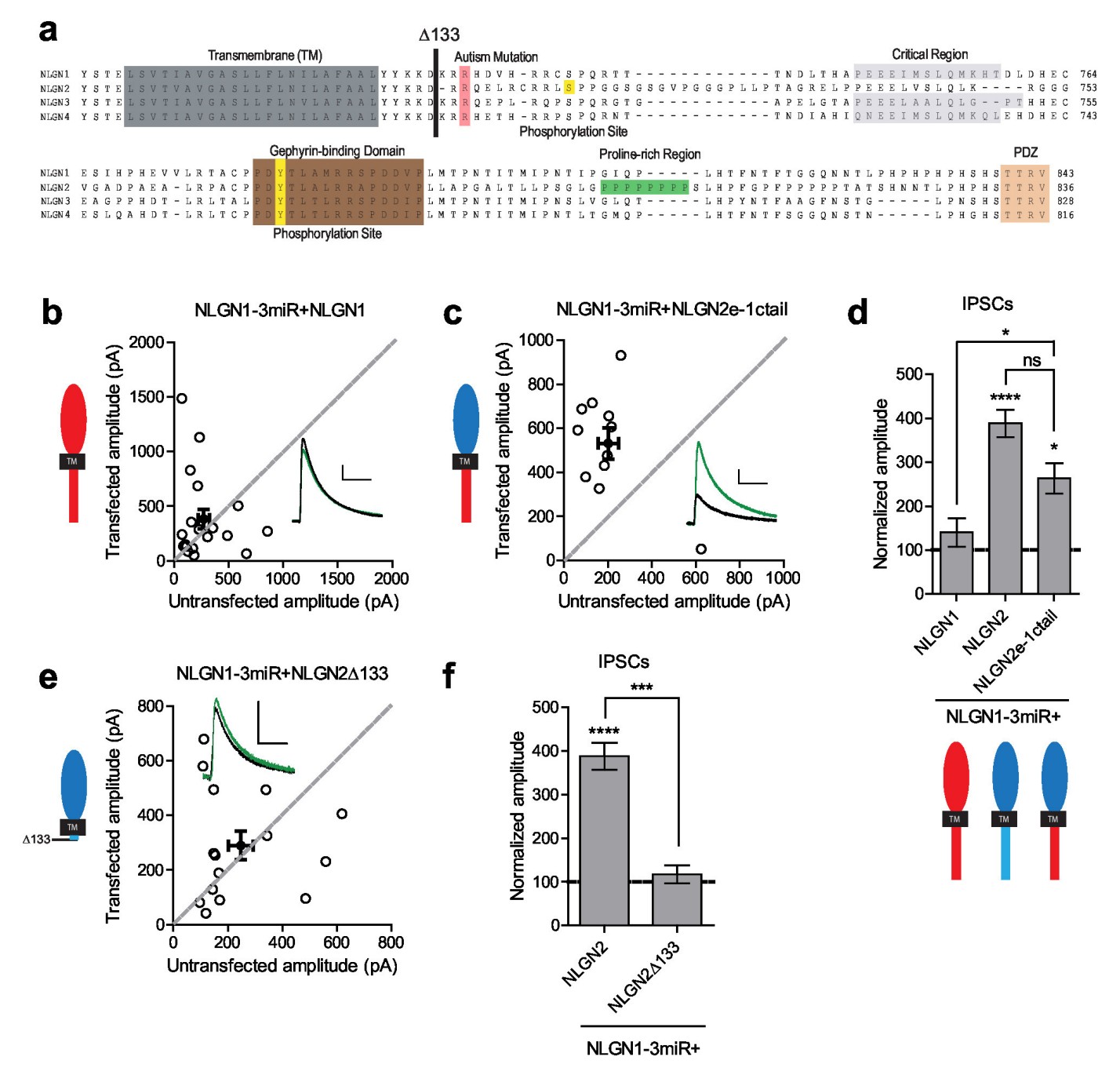

**Figure 4.** While NLGN c-tails do not confer specialization to inhibitory synapses, a c-tail is required for NLGN function. (a) Alignment of NLGN1-4. Sequences correspond to mouse NLGN1, rat NLGN2, and human NLGN3 and NLGN4. Indicated at residue 133 is where most proximal c-tail truncation was performed. Highlighted in gray is the transmembrane domain. In light red is a conserved residue that is the site of an autism-associated mutation in NLGN4. Highlighted in light purple is the critical region, previously shown to be important for NLGN function at excitatory synapses but not inhibitory synapses (*Shipman et al., 2011*). Highlighted in brown is the gephyrin-binding domain. In green is the proline-rich region, and in light orange is the PDZ domain. Phosphorylation sites are highlighted in yellow. (b) Scatter plot showing expression of full-length NLGN1 on a NLGN1-3miR knockdown background failed to enhance inhibitory responses (p=0.6726, n = 19), (c) Scatter plot showing expression of a chimeric NLGN2 extracellular region with NLGN1 c-tail on a NLGN1-3miR knockdown background is able to enhance inhibitory responses (*p=0.0244, n = 11). (d) Summary graph. Effect of expressing NLGN2e-1ctail is significantly greater than with full-length NLGN1 (*p=0.0314). (e) Scatter plot showing expression of the c-tail truncation mutant NLGN2Δ133 on a reduced endogenous neuroligin background is unable to enhance inhibitory synaptic responses, indicating that a c-tail is required for NLGN function (p=0.6387, n = 15). (f) Summary graph of e. Effect of expressing NLGN2Δ133 is significantly less than with full-length NLGN2 (***p=0.0002). For panels b-c and e, open circles are individual pairs, filled circle is mean ± s.e.m. Black sample traces are control, green are

*Figure 4 continued on next page*

*Figure 4 continued*

transfected. Scale bars represent 100 pA and 50 ms. For panels d and f, graph plots transfected amplitude normalized to control ± s.e.m. Significance above each column represents pairwise comparison between transfected and untransfected cells. See also *Figure 4—figure supplement 1*.

The following figure supplement is available for figure 4:

**Figure supplement 1.** Further characterization of c-tail deletion mutant.

## NLGN function at inhibitory synapses requires two distinct regions in the c-tail

Our results suggested that there might be a previously uncharacterized region of the NLGN c-tail that is critical for its function at inhibitory synapses. We made further truncations starting at the region closest to our most terminal c-tail truncation (NLGN2Δ133) and found that deletion of a 35 amino acid stretch from that point (NLGN2Δ98–133) resulted in the loss of enhancement of inhibitory responses (*Figure 7a and e*). Further refinement identified a domain of 16 amino acids that, when deleted (NLGN2Δ117–133), also showed no enhancement (*Figure 7b and e*).

We next sought to identify any candidate residues that could be important within this domain. We were informed by our previous results that any NLGN c-tail would be able to function at inhibitory synapses and thus looked for residues that were shared among all the NLGNs. We focused on two sites, a residue at R705 linked to an autism mutation previously found in NLGN4 that inhibited its function at excitatory synapses (*Bemben et al., 2015*), and a putative phosphorylation site at S714 previously shown to affect gephyrin binding to NLGN2 (*Antonelli et al., 2014*).

Individual mutations at these sites, either by expressing the phospho-null (NLGN2-S714A) (*Figure 7—figure supplement 1a and d*), phospho-mimic (NLGN2-S714D) (*Figure 7—figure supplement 1b and d*), or the analogous autism point mutation (NLGN2-R705C) (*Figure 7—figure supplement 1c and d*), still enhanced inhibitory responses comparable to full-length NLGN2. However, when we expressed the double point mutant of the autism-associated mutation with the phospho-null mutation (NLGN2-R705C-S714A) (*Figure 7c and e*) the enhancement was dramatically reduced, suggesting that together these residues are critical for NLGN function at inhibitory synapses. Notably, when we tested another double point mutant with a phospho-mimic mutation at S714 instead, (NLGN2-R705C-S714D) (*Figure 7d and e*), we still saw robust enhancement of inhibitory currents, showing that our effect is phosphorylation-dependent.

## Gephyrin-dependent and gephyrin-independent mechanisms

Our finding that there are two distinct requirements in the c-tail for NLGN function at inhibitory synapses could explain why we were unable to see an effect earlier with our gephyrin mutants if there are both gephyrin-dependent and gephyrin-independent mechanisms for NLGN function at inhibitory synapses. To determine whether this was the case, we made two more double mutants, combining either the autism mutation or phospho-null mutation with the point mutation in the gephyrin-binding domain that inhibits gephyrin binding (NLGN2-R705C-Y770A or NLGN2-S714A-Y770A). Since the Y770A mutation inherently blocks the gephyrin-dependent pathway, we wanted to know which of our mutations when combined with this manipulation would reveal the presence of a gephyrin-independent pathway. We expect that if either the R705C or S714A point mutations were in a gephyrin-independent pathway, we would observe reduced enhancement of inhibitory currents when combined with the Y770A point mutation that blocks gephyrin binding.

The combined autism mutation and gephyrin-binding double point mutant (NLGN2-R705C-Y770A) exhibited enhancement comparable to full-length NLGN2 (*Figure 7f and h*), while the combined phospho-null mutation and gephyrin-binding double point mutant (NLGN2-S714A-Y770A) showed significantly reduced enhancement (*Figure 7g and h*). Thus, while the three individual point mutations on their own still exhibited enhancement of inhibitory currents, it was only when we mutated both S714 and R705 or S714 and Y770A did we see impairment, suggesting these mutations to be involved in separate pathways. Also, mutation of both R705 and Y770 still exhibited enhancement of inhibitory currents similar to full-length NLGN2, suggesting that they function in the same pathway. Together, we propose the existence of two pathways: a gephyrin-independent

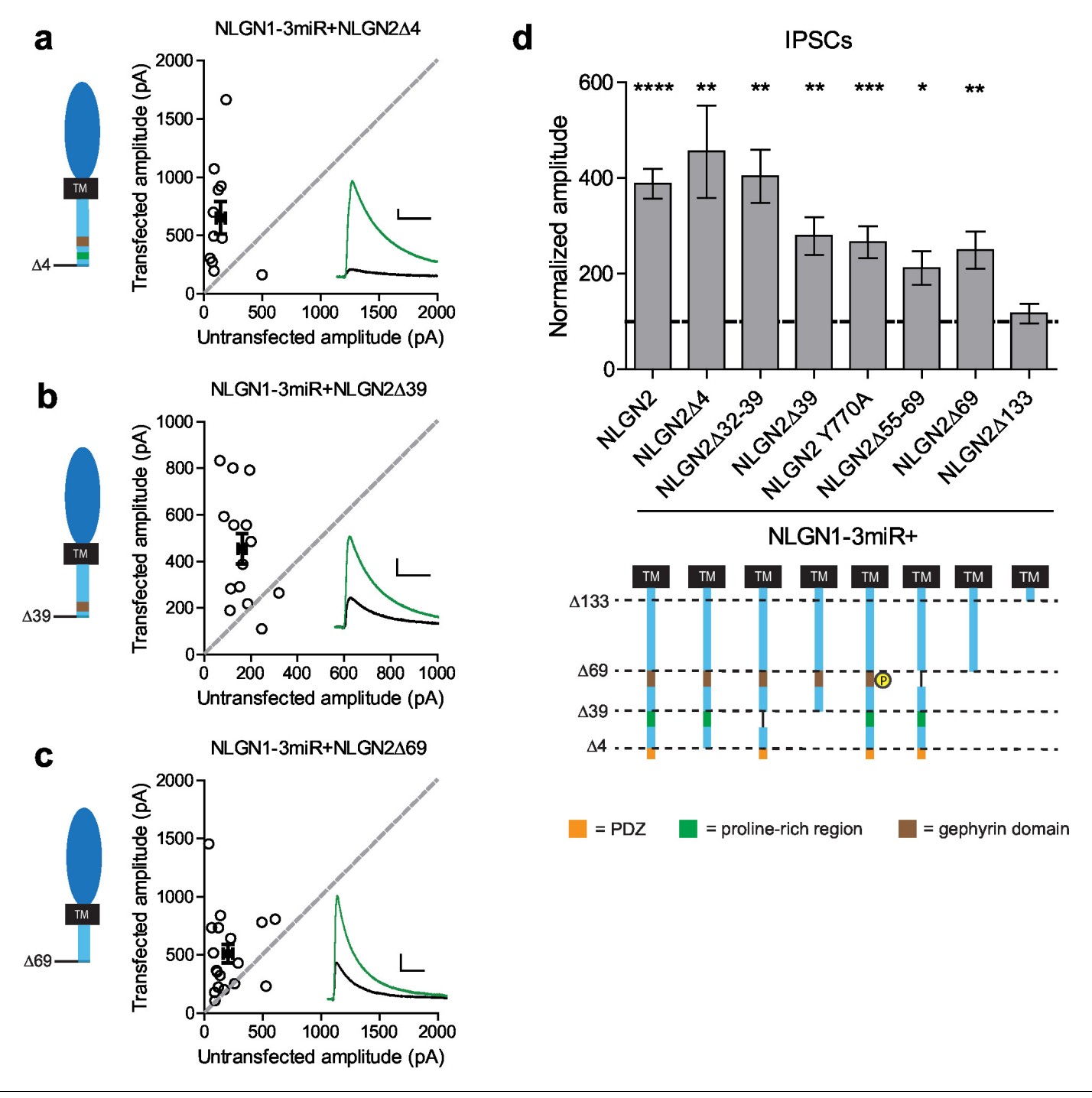

**Figure 5.** Previously identified domains not required for NLGN2 function. (**a–c**) Scatter plots showing replacement of endogenous neuroligin with truncation mutants (**a**) lacking the PDZ domain (NLGN2Δ4) (**p=0.0098, n = 11), (**b**) truncation of all regions downstream of the proline rich region (NLGN2Δ39) (**p=0.0017, n = 14), or (**c**) truncation of all regions downstream of the gephyrin binding domain (NLGN2Δ69) (**p=0.0017, n = 18), is still able to potentiate inhibitory synaptic responses compared to paired neighboring control cells. (**d**) Summary graph. Shown are further mutants including deletion of only the proline rich region (NLGN2Δ32–39) (**p=0.002, n = 10), deletion of only the gephyrin binding domain (NLGN2Δ55–69) (*p=0.0273, n = 10) or expression of a phospho-mimic point mutant that has been shown to disrupt gephyrin binding (NLGN2 Y770A) (***p=0.0005, n = 17), which all still potentiate inhibitory responses. For panels a-c, open circles are individual pairs, filled circle is mean ± s.e.m. Black sample traces are control, green are transfected. Scale bars represent 100 pA and 50 ms. For panel d, graph plots transfected amplitude normalized to control ± s.e.m. Significance above each column represents pairwise comparison between transfected and untransfected cells. See also *Figure 5—figure supplement 1*.

*Figure 5 continued on next page*

*Figure 5 continued*

The following figure supplement is available for figure 5:

**Figure supplement 1.** More detailed NLGN alignment showing where all truncations and point mutations were made.

pathway modulated at S714 and a gephyrin-dependent pathway modulated at both R705 and Y770 (*Figure 8*).

## Discussion

Our results have uncovered the critical features of NLGN2 for proper inhibitory synaptic function and reveal domains in both the extracellular and intracellular regions of the NLGNs that are necessary for their function specifically at inhibitory synapses. Our systematic dissection of the relative contributions of NLGN2 and 3 to inhibitory synaptic function through targeted knockdown of one or the other revealed that while NLGN3 appears to be a dispensable component, NLGN2 is not and that, in fact, NLGN3 function depends on the presence of NLGN2. Further analysis via expression of chimeric NLGN2/3 constructs on a reduced endogenous NLGN background revealed that this difference between NLGN2 and 3 function at inhibitory synapses is due to differences within a domain in the extracellular region corresponding to a site important for neurexin binding, as well as a proximal uncharacterized region.

While this result seemed to suggest that the intracellular region is not important, our results clearly show that, although the intracellular region of any NLGN is sufficient for function at inhibitory synapses, if paired with the extracellular region of NLGN2, a c-tail is very much required. Thus, a NLGN2 truncation lacking most of the intracellular region failed to enhance inhibitory responses while maintaining a normal ability to traffic to the surface. Further refinement identified two distinct regions of the NLGN c-tail corresponding to gephyrin-dependent and gephyrin-independent mechanisms that enable proper function at inhibitory synapses. These results suggest that while the extracellular region is important for specifying function at an inhibitory or excitatory synapse, the intracellular region is critical for carrying out that function at both an inhibitory or excitatory synapse.

Due to the heterogeneous nature of inhibitory synapses, one question that arises is whether our observations are relevant only for particular subsets of inhibitory synapses. Previous work using simultaneous recording from connected neurons showed that NLGN2 knockout animals specifically displayed a deficit in PV but not SOM inputs onto hippocampal pyramidal neurons (*Gibson et al., 2009*). A similar set of studies in NLGN3 knockout animals showed an enhancement of CCK inputs but no change in PV inputs (*Földy et al., 2013*). In our experimental system, we find that while knockdown of NLGN2 critically affects inhibitory transmission, knockdown of NLGN3 does not. Our results are consistent with the observations at PV synapses, suggesting either that our stimulation placement preferentially recruits responses from these types of synapses or that in our system PV inputs are responsible for a large proportion of inhibitory responses.

One limitation of our study is that the results we observe may not occur universally among all brain regions. Indeed, recent work has shown that within the cerebellum the basket/stellate cell synapses onto Purkinje cells exhibit a decrease in mIPSC frequency but not amplitude with knock-out of just NLGN2 while the double knockout of NLGN2 and 3 exhibit a much more dramatic impairment in both mIPSC frequency and amplitude (*Zhang et al., 2015*). In addition, there are inherent limitations with the organotypic slice preparation we have utilized in our experiments including but not limited to the presence of sprouting and rewiring of circuits and the high level of expression imparted by biolistic transfection.

### The role of NLGN3 at inhibitory synapses

What might be the role of NLGN3 at inhibitory synapses since our results show that its absence does not affect inhibitory transmission, and that NLGN3 requires the presence of NLGN2 to function at inhibitory synapses? One possible explanation is that NLGN3 functions as an available mechanism to quickly enhance existing inhibitory synapses. Previous work looking at overexpression of NLGN2

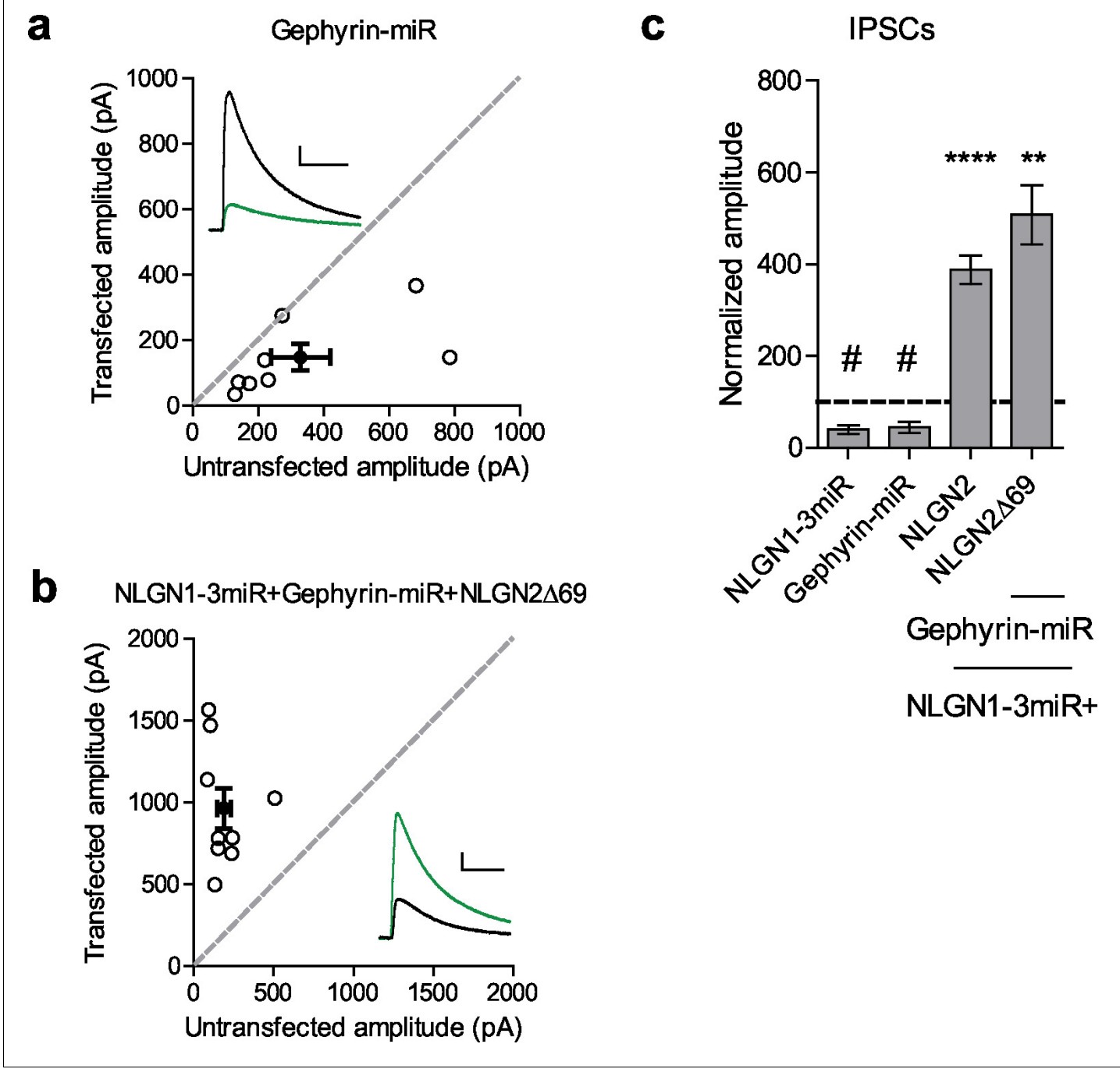

**Figure 6.** Gephyrin-independent NLGN2 function. (a) Scatter plot showing overexpression of a Gephyrin-miR construct to knockdown gephyrin expression significantly reduces inhibitory synaptic responses (#p=0.0156, n = 8). (b) Scatter plot showing replacement of endogenous neuroligin with a truncation mutant lacking all known domains of neuroligin 2 (NLGN2Δ69) on a gephyrin knockdown background is still able to potentiate inhibitory synaptic responses, indicating that neuroligin 2 is able to function independently of gephyrin (**p=0.0039, n = 9). (c) Summary graph. Asterisks indicate significant enhancement compared to control cells where **p<0.01 and ***p<0.001, while hash sign indicates significant depression compared to control cells where #p<0.05. For panels a and b, open circles are individual pairs, filled circle is mean ± s.e.m. Black sample traces are control, green are transfected. Scale bars represent 100 pA and 50 ms. For panel c, graph plots transfected amplitude normalized to control ± s.e.m. Significance above each column represents pairwise comparison between transfected and untransfected cells. See also *Figure 6—figure supplement 1*.

The following figure supplement is available for figure 6:

**Figure supplement 1.** Characterization of Gephyrin knockdown construct.

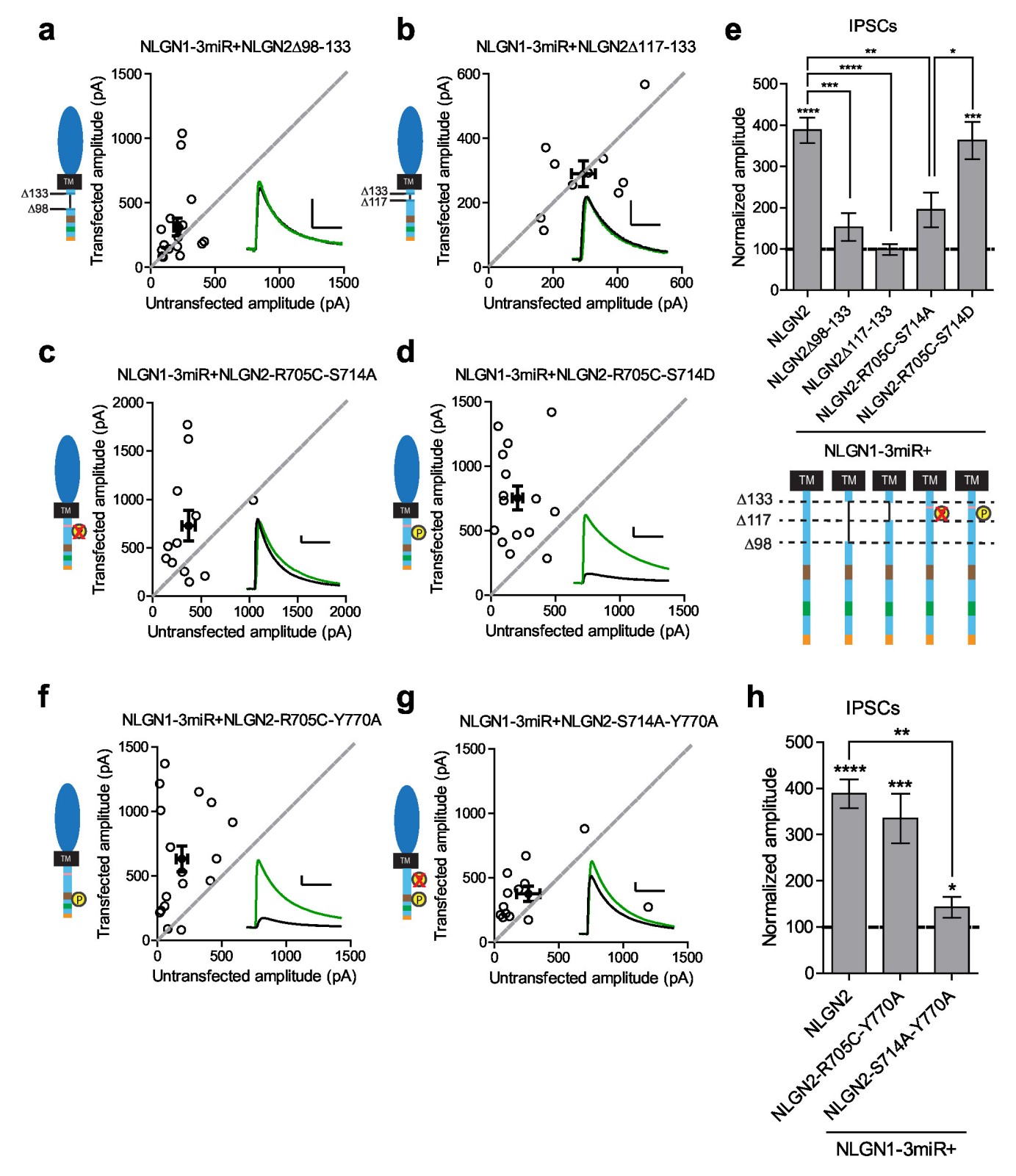

**Figure 7.** Identification of critical residues in NLGN2 c-tail. (**a**) Scatter plot showing deletion of a region from the most proximal c-tail truncation (NLGN2Δ98–133) failed to enhance inhibitory responses (p=0.1698, n = 17). (**b**) Further refining this area of interest, NLGN2Δ117–133 still failed to enhance inhibitory responses (p=0.6953, n = 10). (**c**) Scatter plot showing double point mutant with autism-associated mutation at R705 and phospho-null mutation at S714 (NLGN2-R705C-S714A) also showed deficits in inhibitory transmission (p=0.0522, n = 12). (**d**) Scatter plot showing double point

*Figure 7 continued on next page*

*Figure 7 continued*

mutant with autism-associated mutation at R705 and phospho-mimic mutation at S714 (NLGN2-R705C-S714D) displayed normal enhancement of IPSCs (***p=0.0002, n = 15). (e) Summary graph. Expression of full-length NLGN2 results in greater enhancement of IPSCs compared to NLGN2Δ98–133 (***p=0.0001), NLGN2Δ117–133 (****p<0.0001), and NLGN2-R705C-S714A (**p=0.0061). Expression of NLGN2-R705C-S714D results in greater enhancement of IPSCs compared to NLGN2-R705C-S714A (*p = 0.0429). (f) Scatter plot showing double point mutant with autism-associated mutation at R705 and a phospho-mimic point mutant shown to disrupt gephyrin binding at Y770 is still able to enhance inhibitory responses (***p=0.0005, n = 17). (g) Scatter plot showing double point mutant with phospho-null mutation at S714 and a phospho-mimic point mutant shown to disrupt gephyrin binding at Y770 exhibited modest enhancement of IPSCs (*p=0.0274, n = 13). (h) Summary graph. While the double mutant NLGN2-S714A-Y770A showed enhancement of IPSCs compared to untransfected control cells, the level of potentiation was far less than what we see with full-length NLGN2 (**p=0.0089). For panels a-d, and f-g open circles are individual pairs, filled circle is mean ± s.e.m. Black sample traces are control, green are transfected. Scale bars represent 100 pA and 50 ms. For panel e and h, bar graph plots transfected amplitude normalized to control ± s.e.m. Significance above each column represents pairwise comparison between transfected and untransfected cells. See also *Figure 7—figure supplement 1*.

The following figure supplement is available for figure 7:

**Figure supplement 1.** Individual c-tail point mutations do not affect NLGN2 function.

or 3 in cerebral cortical neurons noticed that the majority of NLGN3 was not localized synaptically while the majority of NLGN2 was synaptically located (*Fekete et al., 2015*). From this, we can surmise the presence of a readily available pool of NLGN3 located outside the synapse that could be recruited and associate with NLGN2 in response to induction of plasticity.

This hypothesis is further strengthened by our result showing that transplanting a domain of the NLGN2 extracellular region corresponding to a key area associated with neurexin binding fully enabled NLGN3 to function at inhibitory synapses by itself. It suggests that NLGN3 is unable to bind to the corresponding neurexin partner at inhibitory synapses alone and that it needs to heterodimerize with NLGN2 in order to function. Previous work has already shown that different neurexin isoforms are expressed at different types of synapses, and that the neurexin isoform expressed at inhibitory synapses can bind to NLGN2 but not NLGN1 (*Fuccillo et al., 2015*; *Futai et al., 2013*). Thus, it is possible that NLGN3, which seems to be able to bind to the neurexins at excitatory synapses by itself, needs to associate with NLGN2 in order to bind to the neurexins at an inhibitory synapse, and this special property of NLGN2 resides in a unique domain in its extracellular region that we have now identified. An alternative hypothesis, and one that does not necessarily require the presence of NLGN2/3 heteromers, is that there is another molecule in addition to neurexin that interacts within that domain of NLGN2 to stabilize the synapse and enable recruitment of NLGN3.

## Extracellular region confers specificity

Our results showing that a chimeric NLGN3 extracellular region with a NLGN2 intracellular region was unable to potentiate inhibitory responses but could still enhance excitatory responses indicates that there are differential requirements for NLGN function at excitatory versus inhibitory synapses and that the extracellular region is critical for mediating this specificity. This is not surprising given that neurexin binding occurs at the extracellular region and that different neurexin isoforms are present preferentially at particular synapses. However, there could be other possible interactors that help define this functional specificity. Previous work has shown that NMDA receptors are able to bind to the extracellular region of NLGN1 but not NLGN2 or 3 (*Budreck et al., 2013*). In addition, the presence of the B splice site insertion in NLGN1 enables it to enhance NMDA currents and promote LTP, and this site is notably absent in NLGN3 (*Shipman and Nicoll, 2012a*). For inhibitory synapses, previous work has shown that MAM domain-containing glycosylphosphatidylinositol anchor proteins (MDGAs) are able to negatively regulate the function of NLGN2 through an interaction within their extracellular region (*Pettem et al., 2013*; *Lee et al., 2013*). Also, there are multiple glycosylation sites in the extracellular region of the NLGNs and previous work has shown that glycosylation state can affect neurexin binding (*Comoletti et al., 2003*; *Hoffman et al., 2004*). However, the analogous glycosylation sites initially found in NLGN1 are conserved among both NLGN2 and 3 within our critical domain, suggesting that they may not explain the difference we see in function. Previously identified cysteine residues that form disulfide bonds within the extracellular region are

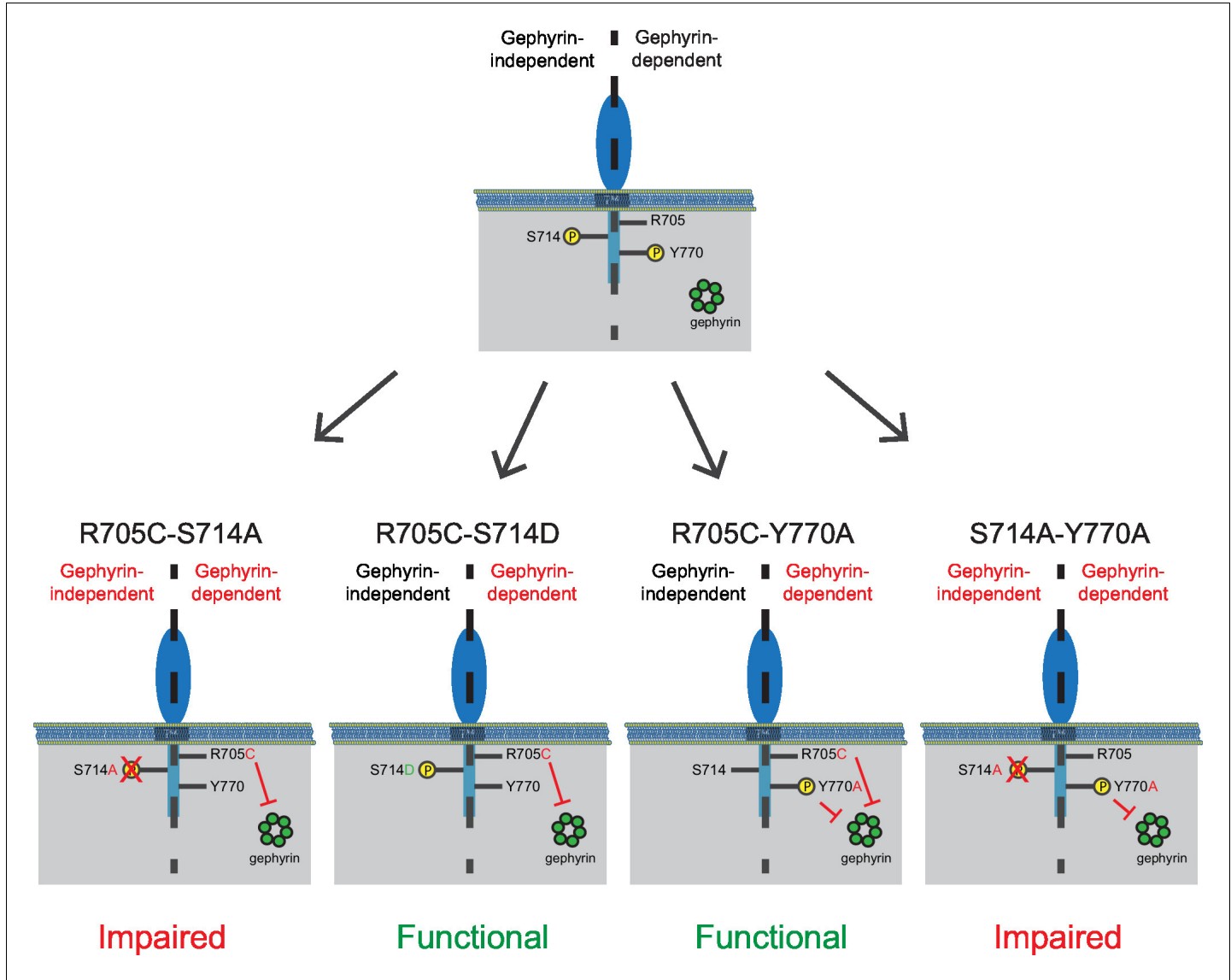

**Figure 8.** Separate gephyrin-dependent and gephyrin-independent mechanisms for neuroligin function at inhibitory synapses. We propose separate gephyrin-dependent and gephyrin-independent mechanisms for NLGN function at inhibitory synapses. In our NLGN2-R705C-S714A manipulation, we block the gephyrin-independent pathway by preventing phosphorylation at the S714 residue, and also block the gephyrin-dependent pathway by the autism-associated mutation at R705. Therefore, the resulting effect we see is impaired function at inhibitory synapses. In our NLGN2-R705C-S714D manipulation, we facilitate the gephyrin-independent pathway by mimicking phosphorylation at the S714 residue, while the gephyrin-dependent pathway is blocked by the autism-associated mutation at R705. The resulting effect we see is intact function at inhibitory synapses. In our NLGN2-R705C-Y770A manipulation, we only block the gephyrin-dependent pathway with the autism-associated mutation at R705 and a mutation at Y770 that blocks gephyrin-binding. The resulting effect we see is intact function at inhibitory synapses. In our NLGN2-S714A-Y770A manipulation, we block the gephyrin-independent pathway by preventing phosphorylation at the S714 residue, and also block the gephyrin-dependent pathway with the mutation at Y770 that blocks gephyrin-binding. Therefore, the resulting effect we see is impaired function at inhibitory synapses. Green is gephyrin. Yellow residues are phosphorylation sites. The Y770A point mutant has been previously characterized and shown to function as a phospho-mimic mutation that blocks gephyrin binding (*Giannone et al., 2013*).

also conserved between NLGN2 and 3 within our critical domain (*Hoffman et al., 2004*). Thus, it would be of interest to determine what molecular interactions are occurring at our critical extracellular domain to mediate NLGN function at inhibitory synapses both in baseline conditions and in response to plasticity.

## Intracellular region contains the toolbox for NLGN function at both excitatory and inhibitory synapses

Our results with chimeric constructs show that any NLGN c-tail is able to function at inhibitory synapses if linked with a NLGN2 extracellular region, and that the NLGN2 c-tail is able to function at excitatory synapses when linked with the NLGN3 extracellular region. This suggests that there is no specificity within the c-tail itself to determine NLGN function at either an excitatory or inhibitory synapse. These results are consistent with previous work showing that overexpression on a wild-type background of chimeric constructs containing NLGN1 extracellular linked to NLGN2 intracellular, or NLGN2 extracellular linked to NLGN1 intracellular, is fully capable of enhancing excitatory and inhibitory currents respectively (*Futai et al., 2013*). Our work eliminates the possibility that previous observations might be due to heterodimerization of chimeric constructs with endogenous NLGNs by expressing all of our chimera on a reduced endogenous NLGN background.

While the c-tail is the toolbox that contains all the functional residues necessary to perform at either an excitatory or inhibitory synapse, it is important to note that the particular type of synapse determines which residues will be utilized. Indeed, prior work has shown that phosphorylation at the gephyrin binding domain enables preferential recruitment of PSD-95 versus gephyrin to the intracellular region of NLGN1 (*Giannone et al., 2013*). In addition, while it has been shown that there is a critical residue in the NLGN3 c-tail that is important for its function at excitatory synapses, mutation of the analogous residue in NLGN2 does not affect its function at inhibitory synapses (*Shipman et al., 2011*). Therefore, the mechanisms we describe are specific for NLGN function at inhibitory synapses.

## Gephyrin-dependent and gephyrin-independent mechanisms for NLGN function at inhibitory synapses

We were surprised to find that not only the PDZ and collybistin domain are dispensable for NLGN function at inhibitory synapses but also the gephyrin-binding domain as well. There have been previous reports of gephyrin-independent function at inhibitory synapses, either in development or during plasticity (*Danglot et al., 2003*; *Niwa et al., 2012*). In addition, previous work done in mice lacking the cytoplasmic protein dystrophin showed selective reduction in clustering of α2 containing GABA$_A$ receptors but no change in gephyrin clustering, suggesting there might be dystrophin-dependent and gephyrin-independent mechanisms for clustering of specific GABA$_A$ receptor subtypes (*Knuesel et al., 1999*). Thus, it would be of interest to determine whether phosphorylation of the S714 site affects dystrophin function.

Phosphorylation at the S714 residue of NLGN2 has been found to negatively regulate the interaction between NLGN2 and gephyrin (*Antonelli et al., 2014*). However, our results indicate that phosphorylation at this site positively regulates NLGN2 function through a gephyrin-independent mechanism, since only the phospho-null form is able to block enhancement of inhibitory currents when paired with a mutation that blocks the gephyrin-dependent pathway (*Figure 8*). This discrepancy might be explained by the previous study's use of non-neuronal cells for characterization of the interaction between gephryin and the phosphorylation state at that site. It remains to be determined what kinase phosphorylates this particular site and what molecules are recruited in response to phosphorylation to mediate gephyrin-independent NLGN2 function.

The R705C point mutation is the only autism-associated mutation in the NLGNs identified to date that resides in the intracellular region. Previous characterization of this autism mutation at the analogous residue in NLGN3 showed no effect at inhibitory synapses (*Etherton et al., 2011*). This is consistent with our results showing that NLGN2 with the R705C point mutation alone is fully capable of enhancing inhibitory transmission (*Figure 7—figure supplement 1c and d*). However, by combining the R705C mutation with the S714A mutation, we were able to uncover a deficit in NLGN function (*Figure 8*). Similarly, it was only by combining the Y770A mutation with the S714A mutation, not the R705C mutation, that we were able to prevent NLGN2 from enhancing inhibitory currents, suggesting that the R705C mutation operates via a gephyrin-dependent pathway (*Figure 8*). These results underscore the additive effects of multiple point mutations, and highlight how disease phenotypes can be due to the interaction of a number of different factors. While the R705C mutation was shown to inhibit PKC-mediated interaction with NLGN4X at excitatory synapses, there is no analogous PKC phosphorylation site in NLGN2 (*Bemben et al., 2015*). Thus, it remains to be determined what

molecules interact at this site and how this autism-associated mutation results in disruption of gephyrin binding to NLGN2.

Our findings have highlighted the separate roles of the NLGN extracellular and intracellular regions. The extracellular region is important for specifying function at a particular synapse while the intracellular region is important for carrying out the molecular mechanisms at either an excitatory or inhibitory synapse. We identify a domain in the extracellular region that is necessary and sufficient for specifying NLGN function at inhibitory synapses and elucidate both a gephyrin-dependent and gephyrin-independent mechanism for NLGN function at inhibitory synapses operating via intracellular interactions modulated by a phosphorylation site and an autism-associated mutation. These results advance our fundamental understanding of NLGN function at inhibitory synapses and provide multiple new avenues of study for further dissection of the various molecular interactors of the NLGNs, both extracellularly and intracellularly, and in normal and disease conditions.

## Materials and methods

### Experimental constructs

Neuroligin overexpression constructs have previously been described (*Shipman et al., 2011*) and were based on RNAi-proofed HA-tagged rat NLGN2 and human NLGN3. We used an RNAi-proof NLGN1 isoform containing the B splice site. All neuroligin constructs were cloned into the pCAGGS expression plasmid by either PCR for truncations or overlap extension PCR for chimera, deletions, and point mutations followed by In-Fusion cloning (Takara Bio, Japan). All NLGN2 constructs contained an HA-tag and all chimera retained the NLGN2 version of the transmembrane domain. The triple microRNA construct to knockdown neuroligins 1, 2, and 3 (NLGN1-3miR) and NLGN3miR have been previously characterized (*Shipman and Nicoll, 2012a*; *Shipman et al., 2011*). Targeting sequence for the NLGN2miR construct was TTGCTGTTGAACTTGCTCCAT. Gephyrin-miR targeting sequence was AACAGGGAATGAGCTACTAAA. All targeting sequences were cloned using BLOCK-iT miR RNAi kit (Invitrogen, Waltham, MA). All constructs used for biolistic transfection and lentiviral production co-expressed either a GFP or mCherry fluorophore for visualization.

### Lentivirus production

HEK293T cells were co-transfected with psPAX2, pVSV-G, and either NL2miR or Gephyrin-miR using FuGENE HD (Promega, Madison, WI). Supernatant was collected 40 hr later, filtered, and concentrated using PEG-it Virus Precipitation Solution (System Biosciences, Palo Alto, CA). Resulting pellet was resuspended in Opti-mem, flash-frozen, and stored at –80°C.

### qRT-PCR

Primary rat hippocampal dissociated neurons were prepared at E18.5 and infected with lentivirus expressing a Gephyrin-miR, NLGN2miR, or control GFP construct at DIV 4–7. Neurons were harvested at DIV17-18 by lysis and reverse transcribed to synthesize cDNA using a Power SYBR Green Cells-to-CT kit (Life Technologies, Waltham, MA). Amplification of cDNA by real-time PCR was quantified using SYBR Green with the following sequence specific primers: Neuroligin 2, forward-CATTCAGAAGGGCTGTTCCA, reverse- GTCTTCCCGGGGAGCTAGTAG; Gephryin, forward- GGGAATGAGCTACTAAATCCTG, reverse- TGATACCCTCATTCAAGGCA.

### Immunoblotting

Primary rat hippocampal dissociated neurons were prepared at E18.5 and infected with lentivirus expressing Gephyrin-miR construct at DIV 4–7. Neurons were harvested at DIV17-18 in Tris-buffered saline (25 mM Tris pH 7.4, 150 mM NaCl) plus 0.5% Triton-X and protease inhibitor mix (Roche Applied Sciences, cOmplete Protease Inhibitor Cocktail Tablets, Germany). After incubation at 4°C for 30 min, cell lysates were centrifuged for 15 min at 12000 g. Proteins were resolved by SDS-PAGE and analyzed by western blot using antibodies against gephyrin (1:5000, Synaptic Systems, Germany) and actin (1:5000, Millipore C4, Billerica, MA).

## Immunostaining and Imaging

Primary rat hippocampal dissociated neurons were prepared at E18.5 and transfected using Lipofectamine 2000 with pCaggs NLmiRs-GFP and either pCaggs-NLGN2 or pCaggs-NLGN2Δ133. 7 days later cells were prepped for immunostaining for surface and intracellular markers. Cells were first washed with cold HEPES external buffer (115 mM NaCl, 3.5 mM KCl, 10 mM HEPES, 20 mM Glucose, 1 mM $MgCl_2$, 1.5 mM $CaCl_2$, adjusted to pH 7.3 with NaOH) then incubated with anti-HA antibody (Y-11, 1:250, Santa Cruz Biotech) for 20 mins in the cold. Cells were then washed with HEPES external buffer and fixed in 4% paraformaldehyde in 4% PBS for 15 min room temperature. Cells were then washed with PBS and blocked in 10% goat serum in PBS with 0.1% Triton X-100. Cells were then washed with PBS with 0.1% Triton X-100 and labeled with Alexa 546- conjugated anti-rabbit secondary antibody (1:250, Thermo-Fisher Scientific, Waltham, MA) in PBS with 0.1% Triton X-100 with 2.5% goat serum and mounted with SlowFade Gold Antifade Mountant (Molecular Probes, Waltham, MA). Neurons were imaged with a 100x objective on a Zeiss LSM 510 confocal microscope. For analysis, images were collected and quantified by normalizing the fluorescence intensity of surface-expressed NLGN2 (or NLGN2Δ133) with the fluorescence intensity of GFP in each cell using ImageJ (NIH, Bethesda, MD). All data analysis was done blinded to experimental conditions.

## Slice culture and biolistic transfection

Rat slice cultures were prepared on P6–8 as previously described (*Stoppini et al., 1991*). All experiments were performed in accordance with established protocols approved by the University of California San Francisco Institutional Animal Care and Use Committee.

Sparse biolistic transfections of organotypic slice cultures were performed 1 day after culturing as previously described (*Schnell et al., 2002*). Briefly, 100 µg total of mixed plasmid DNA was coated on 1 µm-diameter gold particles in 0.5 mM spermidine, precipitated with 0.1 mM $CaCl_2$, and washed four times in pure ethanol. The gold particles were coated onto PVC tubing, dried using ultra-pure $N_2$ gas, and stored at 4°C in desiccant. DNA-coated gold particles were delivered with a Helios Gene Gun (BioRad, Hercules, CA). When biolistically expressing two plasmids, gold particles were coated with equal amounts of each plasmid and plasmids always expressed different fluorescent markers. Observed frequency of coexpression was nearly 100%. Slices were maintained at 34°C with media changes every other day.

## Electrophysiological recording

Recordings were performed at 7–10 DIV after 6–9 days of expression. Dual whole-cell recordings of CA1 pyramidal neurons were done by simultaneously recording responses from a fluorescent transfected neuron and a neighboring untransfected control neuron. Synaptic responses were evoked by stimulating with a monopolar glass electrode filled with aCSF in stratum radiatum of CA1. Typically each pair of neurons is from a separate slice, whereas on rare occasions two pairs may come from one slice. For all paired recordings, the number of experiments (*n*) reported in the figure legends refer to the number of pairs. Pyramidal neurons were identified by morphology and location. To ensure stable recording, membrane holding current, input resistance, and pipette series resistance were monitored throughout recording. All recordings were made at 20–25°C using glass patch electrodes filled with an internal solution consisting of 135 mM $CsMeSO_4$, 8 mM NaCl, 10 mM HEPES, 0.3 mM EGTA, 4 mM Mg-ATP, 0.3 mM Na-GTP, 5 mM QX-314, and 0.1 mM spermine and an external solution containing 119 mM NaCl, 2.5 mM KCl, 4 mM $MgSO_4$, 4 mM $CaCl_2$, 1 mM $NaH_2PO_4$, 26.2 mM $NaHCO_3$ and 11 mM glucose bubbled continuously with 95% $O_2$ and 5% $CO_2$. Recordings of IPSCs were made in the presence of D-APV (100 µM, Cayman Chemical, Ann Arbor, MI) and NBQX (10 µM, Abcam, UK) to block NMDA and AMPA-mediated currents respectively. Recordings of excitatory current were made in the presence of picrotoxin (100 µM, Spectrum Chemical, Gardena, CA) to block inhibitory currents and a small (50 nM) amount of NBQX to reduce epileptiform activity at −70 mV (AMPA). AMPAR-mediated currents were measured at the peak of the current.

## Statistical analysis

All paired whole-cell data were analysed using a two-tailed Wilcoxon matched-pairs signed rank test. For comparisons between different experimental groups, a Mann Whitney test was used on the

ratios of the transfected cell to the control cell. Data analysis was carried out in Igor Pro (Wavemetrics, Portland, OR), GraphPad Prism (GraphPad Software, La Jolla, CA) and Excel (Microsoft, Redmond, WA).

## Acknowledgements

We thank S Shipman for initial conception of this project and helpful comments on the manuscript. We thank K Bjorgan and M Cerpas for technical assistance and members of the Nicoll laboratory for thoughtful comments and suggestions. We thank M Bemben for technical assistance on review experiments. Q-AN was supported by a Predoctoral Fellowship from the American Heart Association. MEH was supported by a National Science Foundation Graduate Research Fellowship. RAN is supported by grants from the National Institute of Mental Health.

## Additional information

### Funding

| Funder | Grant reference number | Author |
| --- | --- | --- |
| American Heart Association | Predoctoral Fellowship | Quynh-Anh Nguyen |
| National Science Foundation | Graduate Research Fellowship | Meryl E Horn |
| National Institute of Mental Health | | Roger A Nicoll |

The funders had no role in study design, data collection and interpretation, or the decision to submit the work for publication.

### Author contributions

Q-AN, Designed experiments, Performed experiments and analysed and interpreted data, Wrote the manuscript; MEH, Designed experiments, Performed experiments and analysed and interpreted data, Provided manuscript comments and directed revisions; RAN, Designed experiments, Helped interpret data and supervised the project, Provided manuscript comments and directed revisions

### Author ORCIDs

Roger A Nicoll, http://orcid.org/0000-0002-6977-4632

### Ethics

Animal experimentation: This study was performed in strict accordance with the recommendations in the Guide for the Care and Use of Laboratory Animals of the National Institutes of Health. All experiments were performed in accordance with established protocols approved by the University of California San Francisco Institutional Animal Care and Use Committee (PHS Assurance #A3400-01).

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
