## [Decision Letter]

Thank you for submitting your article "Distinct roles for extracellular and intracellular domains in neuroligin function at inhibitory synapses" for consideration by *eLife*. Your article has been favorably evaluated by Gary Westbrook (Senior Editor) and three reviewers, one of whom is a member of our Board of Reviewing Editors. The reviewers have discussed the reviews with one another and the Reviewing Editor has drafted this decision to help you prepare a revised submission.

Summary:

Nguyen et al. report the roles of distinct extracellular and intracellular domains of neuroligins (NLGNs) at inhibitory synapses. The authors first find that NLGN2 but not NLGN3 is required for inhibitory synaptic transmission in cultured neurons. In follow-up experiments, the authors mainly use molecular replacement approaches to map out critical regions associated with NLGN2-dependent regulation of inhibitory transmission. The authors identify two domains in the extracellular region of NLGN2 (splice A site domain and an uncharacterized adjacent domain) that confer NLGN function at inhibitory synapses. Intracellularly, the authors find that a c-tail from NLGN1/2/3 is required for NLGN function at inhibitory synapses. Furthermore, they find that previously identified cytoplasmic domains including the PDZ-binding motif and gephyrin-binding domain are not required for NLGN2 function, but that an upstream region (δ 117-133) and two residues within this region, one autism-related (R705) and the other phosphorylatable (S714), are important for NLGN2 function. Interestingly, the authors find that these residues together with a residue in the gephyrin-binding domain (Y770) are collectively important for NLGN2 function; combinations of residues both in the gephyrin-dependent and -independent pathways are important to fully support NLGN2 function. This is an interesting, well-performed and thorough study, and a well-written manuscript that is appropriate for *eLife*. The findings provide novel insight into the relevance and function of previously unidentified regions of Nlgn at inhibitory synapses.

Essential revisions:

1) This study used many constructs carrying deletions and mutations of NLGN2 and NLGN3. Ideally, their total and surface expression levels should be tested using biochemical (i.e. biotinylation) or cell biological methods, although some were tested by surface staining (Figure 4—figure supplement 1). The extracellular deletion variants were verified of their support for excitatory synaptic transmission (Figure 3—figure supplement 1), which is admittedly a nicer way, but most of the cytoplasmic variants of NLGN2 were not tested for total/surface expression.

2) The authors identify the proximal part of the extracellular region of Nlgn2 as required for the functional differences between Nlgn2 and Nlgn3 (Figure 3). The study would be enhanced if the authors could identify the property that makes this proximal region unique. The authors could use binding assays to test if this proximal region confers binding properties to Nlgn3 that allow binding of this Nlgn3 to neurexins expressed at inhibitory synapses. They could also use similar binding assays to test whether this proximal region might be required to allow Nlgn3 to associate with Nlgn2 in heteromers at inhibitory synapses. Even if the results are negative it would still be informative, as this would strengthen their alternative hypothesis that there could be another molecule interacting with the Nlgn2 proximal domain (subsection “Extracellular region confers specificity”).

3) A challenge in dissecting the contribution of the Nlgn c-tail to inhibitory synaptic function is that the downstream signaling pathways are largely unknown. Although their experiments suggest that the R705C mutation acts via a gephyrin-dependent pathway, the molecular basis of this is less clear. The authors previously showed that in Nlgn4, this mutation impairs phosphorylation of neighboring T707 by PKC (Bemben et al., PNAS 2015). T707 is absent in Nlgn1-3, suggesting an alternative mechanism. Biochemical analysis of gephryin binding in R705C mutants would help to clarify how this mutation might affect the gephyrin-dependent pathway. In these experiments, the R705C mutant can be compared to S714A mutant, which would not be expected to affect gephyrin binding. In addition, is it possible to check that the phosphorylation of neighboring S714 is not affected by the R705C mutation?

4) The authors propose that S714 modulates a gephyrin-independent pathway, based on their findings that the S714A single mutation does not impair inhibitory responses, whereas the S714A-Y770A double mutant does. The authors could functionally test the effect of the single S714A mutation by expressing the single Nlgn2-S714A mutant in an Nlgn1-3 miR+/gephyrin-miR+ background, which would be expected to fail to enhance inhibitory responses.

5) The main approach employed in the current study is electrophysiological recordings of rat organotypic slice cultured neurons. To corroborate the key findings, some of the phenotypes (if not all) should be confirmed by independent immunocytochemical approaches.

---

## [Author Response]

*[…] Essential revisions:*

1) This study used many constructs carrying deletions and mutations of NLGN2 and NLGN3. Ideally, their total and surface expression levels should be tested using biochemical (i.e. biotinylation) or cell biological methods, although some were tested by surface staining (Figure 4—figure supplement 1). The extracellular deletion variants were verified of their support for excitatory synaptic transmission (Figure 3—figure supplement 1), which is admittedly a nicer way, but most of the cytoplasmic variants of NLGN2 were not tested for total/surface expression.

We have done surface staining for the HA-tag construct in two of our most critical double point mutations which exhibited impaired inhibitory synaptic transmission (Figure 9). As shown in Figure 9, both the NLGN2-R705C-S714A and NLGN2-S714A-Y770A mutants did not display any impairment in surface trafficking. Actually, they seem to be expressed to a greater extent than wild-type NLGN2 on the surface.

Author response image 1.Dissociated hippocampal rat cultures were prepared at E18.5 and transfected with NLmiRs-GFP and either NLGN2, NLGN2-R705C-S714A, or NLGN2-S714A-Y770A at DIV 8.After 1 week cells were stained for surface HA (Abcam ab9110, 1:200). HA signal was normalized to GFP. (NLGN2 n=5 cells, NLGN2-R705C-S714A n=7 cells, NLGN2-S714A-Y770A n=7 cells; *p=0.0177, **p=0.0025).**DOI:**
http://dx.doi.org/10.7554/eLife.19236.017

2) The authors identify the proximal part of the extracellular region of Nlgn2 as required for the functional differences between Nlgn2 and Nlgn3 (Figure 3). The study would be enhanced if the authors could identify the property that makes this proximal region unique. The authors could use binding assays to test if this proximal region confers binding properties to Nlgn3 that allow binding of this Nlgn3 to neurexins expressed at inhibitory synapses. They could also use similar binding assays to test whether this proximal region might be required to allow Nlgn3 to associate with Nlgn2 in heteromers at inhibitory synapses. Even if the results are negative it would still be informative, as this would strengthen their alternative hypothesis that there could be another molecule interacting with the Nlgn2 proximal domain (subsection “Extracellular region confers specificity”).

There are multiple hurdles underlying the suggested experiment to express different variants of neurexins to test binding of NLGN3. First, there is no readily available library of all the neurexin variants expressed at inhibitory synapses from which we can obtain constructs to overexpress. Even if there were, there are a plethora of neurexin variants and combinations, including splice isoforms, which could be possibly expressed, making it unwieldy to perform this experiment.

However, we have attempted to characterize our critical extracellular domain further by looking at the crystal structure of NLGN2 which has previously been resolved [1]. We aligned the structure of NLGN2 (3BL8) to the crystal structure of the NLGN1/Neurexin1β complex [2] (3BIW) (Figure 10), to approximate the location at which neurexin would bind to NLGN2. Highlighted in red is our critical extracellular domain, which appears to be on the side of the neuroligin away from the neurexin-binding interface, suggesting that molecular partners other than neurexins underlie the ability of this domain to confer neuroligin function at inhibitory synapses. Indeed, there are a number of candidate molecules hypothesized to associate with NLGN2 [3] and it would be of interest for future studies to screen through possible interactors for binding to our identified domain. One caveat of this structural prediction is that the neurexin binding sites are based on the NLGN1/Neurexin1β complex and it is possible that neurexins at inhibitory synapses bind to NLGN2 at a different site. In addition, we noticed that the published crystal structure of NLGN2 is missing 14 amino acids around and within the splice site A site, which could affect the true structure of the protein.

We were unable to characterize and align NLGN3 to our structural analysis due to the lack of a crystal structure and the inability of structure prediction software to fully predict the structure of our protein due to the highly disordered region around splice site A.

Author response image 2.Crystal structure of NLGN2 dimer (in blue) aligned with Neurexin1β structure (in orange).In red is the critical extracellular domain in NLGN2 identified in our study.**DOI:**
http://dx.doi.org/10.7554/eLife.19236.018

We do not think that our identified critical domain in the proximal region is required to allow NLGN3 to associate with NLGN2 in heteromers since wild-type NLGN3 already associates with NLGN2, as shown in the blot (Figure 11). In this experiment, HEK cells were transfected with constructs for either GFP alone, GFP+NLGN2, GFP+NLGN3, or both NLGN2 and 3. The NLGN2 construct contained an HA tag that we immunoprecipitated for and then blotted for NLGN3 to determine if the two bind. We see robust NLGN3 signal in the IP samples from cells expressing both NLGN2 and NLGN3. We can also see that we did enrich for HA-associated protein complexes since the NLGN3 signal in the input lane from cells expressing GFP+ NLGN3 goes away in the IP lane. We see some signal in the GFP+NLGN2 samples in the IP lane and this is due to the fact that the antibody we use can also detect other neuroligins when they are highly overexpressed.

Author response image 3.HEK293T cells were transfected with either GFP, GFP+NLGN2, GFP+NLGN3, or NLGN2+NLGN3.After 2 days lysates were harvested and incubated in HA-conjugated agarose beads (Sigma-Aldrich, Saint Louis, MO) to immunoprecipitate HA-tagged protein complexes. Samples were run on a 4–12% Bis-Tris gel and probed for HA (Santa Cruz 1:2000), NLGN3 (Synaptic Systems 1:1000), and actin (Millipore C4 1:5000). Input samples are 3% of IP samples. Blots are representative of at least 2 experimental and technical replicates. Size indicated on left in kDa.**DOI:**
http://dx.doi.org/10.7554/eLife.19236.019

*3) A challenge in dissecting the contribution of the Nlgn c-tail to inhibitory synaptic function is that the downstream signaling pathways are largely unknown. Although their experiments suggest that the R705C mutation acts via a gephyrin-dependent pathway, the molecular basis of this is less clear. The authors previously showed that in Nlgn4, this mutation impairs phosphorylation of neighboring T707 by PKC (Bemben et al., PNAS 2015). T707 is absent in Nlgn1-3, suggesting an alternative mechanism. Biochemical analysis of gephryin binding in R705C mutants would help to clarify how this mutation might affect the gephyrin-dependent pathway. In these experiments, the R705C mutant can be compared to S714A mutant, which would not be expected to affect gephyrin binding. In addition, is it possible to check that the phosphorylation of neighboring S714 is not affected by the R705C mutation?*

We transfected HEK cells with gephyrin and either NLGN2, NLGN2-R705C, or NLGN2-S714A. We then immunoprecipitated for the HA-tagged NLGN2 and blotted for gephyrin to see if we could detect any differences in gephyrin binding to the different point mutants. As shown (Figure 12), we could not detect any noticeable difference in the gephyrin signal between the point mutants. This result is not definitive in showing that there is no effect on gephyrin interactions with our point mutants due to the fact that we are highly overexpressing them and thus it is possible that there are slight nuances in the interaction that we are not able to pick up in this assay. In addition, we are using non-neuronal cells which could impact our results. While it would be ideal to use neuronal cells for this assay, we would be hampered by the fact that the gephyrin-NLGN2 interaction itself is quite weak. As shown in our blots, we see robust gephyrin signal in our input lanes. However, when we enrich for NLGN2, we see about 90% less signal suggesting that most of the gephyrin being overexpressed is not binding to NLGN2. This signal would be even weaker/undetectable if we were to use neuronal cells and endogenous gephyrin.

Unfortunately, we are not able to check whether phosphorylation of neighboring S714 is affected by the R705C mutation due to lack of a phospho-specific antibody for S714.

Author response image 4.HEK293T cells were transfected with either GFP, NLGN2+gephyrin, NLGN2R705C+gephryin, or NLGN2S714A+gephyrin.After 2 days lysates were harvested and incubated in HA-conjugated agarose beads (Sigma) to immunoprecipitate HA-tagged protein complexes. Samples were run on a 4–12% Bis-Tris gel and probed for HA (Santa Cruz 1:2000), gephyrin (Synaptic Systems 1:5000), and actin (Millipore C4 1:5000). Input samples are 3% of IP samples. Blots are representative of at least 2 technical replicates. Size indicated on left in kDa.**DOI:**
http://dx.doi.org/10.7554/eLife.19236.020

*4) The authors propose that S714 modulates a gephyrin-independent pathway, based on their findings that the S714A single mutation does not impair inhibitory responses, whereas the S714A-Y770A double mutant does. The authors could functionally test the effect of the single S714A mutation by expressing the single Nlgn2-S714A mutant in an Nlgn1-3 miR+/gephyrin-miR+ background, which would be expected to fail to enhance inhibitory responses.*

We thank the reviewer for this suggestion. We have performed this experiment and indeed expression of NLGN2-S714A along with NLGN1-3miR + Gephyrin-miR fails to enhance inhibitory responses.

Author response image 5.Scatterplot and bar graph showing expression of NLGN2S714A along with NLGN1-3miR and Gephyrin-miR fails to enhance inhibitory responses and is significantly different than expression of full-length NLGN2 on the NLGN1-3miR background (n = 8, **p=0.0023, ****p<0.0001).Scale bar represents 25pA and 50ms.**DOI:**
http://dx.doi.org/10.7554/eLife.19236.021

*5) The main approach employed in the current study is electrophysiological recordings of rat organotypic slice cultured neurons. To corroborate the key findings, some of the phenotypes (if not all) should be confirmed by independent immunocytochemical approaches.*

With due respect, for multiple reasons we would argue that immunocytochemical approaches would not significantly enhance the impact of our study. First, previous work in our lab has shown that neurons expressing point mutants which affect the function of a NLGN, as determined by electrophysiological recordings, could display normal presynaptic specializations [4]. In addition, it is difficult to determine the effects our mutants have on postsynaptic specializations since there is no structural approximation of inhibitory synapse number equivalent to the spine density used for excitatory synapses. Indeed, the postsynaptic marker that is used most often for inhibitory synapses is gephyrin, which would not be ideal to use for our purposes.

References:

1) Koehnke, J. et al. Crystal structure of the extracellular cholinesterase-like domain from neuroligin-2. Proc Natl Acad Sci U S A 105, 1873-1878, doi:10.1073/pnas.0711701105 (2008).

2) Arac, D. et al. Structures of neuroligin-1 and the neuroligin-1/neurexin-1 beta complex reveal specific protein-protein and protein-Ca2+ interactions. Neuron 56, 992-1003, doi:10.1016/j.neuron.2007.12.002 (2007).

3) Kang, Y. et al. A combined transgenic proteomic analysis and regulated trafficking of neuroligin-2. J Biol Chem 289, 29350-29364, doi:10.1074/jbc.M114.549279 (2014).

4) Shipman, S. L. & Nicoll, R. A. A subtype-specific function for the extracellular domain of neuroligin 1 in hippocampal LTP. Neuron 76, 309-316, doi:10.1016/j.neuron.2012.07.024 (2012).